# Small molecule degraders of the hepatitis C virus protease reduce susceptibility to resistance mutations

Mélissanne de Wispelaere [1,4], Guangyan Du[2,3,4], Katherine A. Donovan [2,3], Tinghu Zhang[2,3], Nicholas A. Eleuteri[3], Jingting C. Yuan[3], Joann Kalabathula[3], Radosław P. Nowak [2,3], Eric S. Fischer [2,3], Nathanael S. Gray [2,3] & Priscilla L. Yang[1]

Targeted protein degradation is a promising drug development paradigm. Here we leverage this strategy to develop a new class of small molecule antivirals that induce proteasomal degradation of viral proteins. Telaprevir, a reversible-covalent inhibitor that binds to the hepatitis C virus (HCV) protease active site is conjugated to ligands that recruit the CRL4[CRBN] ligase complex, yielding compounds that can both inhibit and induce the degradation of the HCV NS3/4A protease. An optimized degrader, DGY-08-097, potently inhibits HCV in a cellular infection model, and we demonstrate that protein degradation contributes to its antiviral activity. Finally, we show that this new class of antiviral agents can overcome viral variants that confer resistance to traditional enzymatic inhibitors such as telaprevir. Overall, our work provides proof-of-concept that targeted protein degradation may provide a new paradigm for the development of antivirals with superior resistance profiles.

---

[1] Department of Microbiology and Blavatnik Institute, Harvard Medical School, Boston, MA 02115, USA. [2] Department of Biological Chemistry and Molecular Pharmacology, Harvard Medical School, Boston, MA 02115, USA. [3] Department of Cancer Biology, Dana-Farber Cancer Institute, Boston, MA 02215, USA. [4]These authors contributed equally: Mélissanne de Wispelaere, Guangyan Du. Correspondence and requests for materials should be addressed to P.L.Y. (email: priscilla_yang@hms.harvard.edu)

In antiviral research, there is a major need to achieve potent broad-spectrum inhibition of human viral pathogens with high barriers to resistance. One of the traditional antiviral discovery strategies is to develop high-affinity ligands that bind and inhibit the functions of viral proteins, generally enzymes such as viral polymerases and proteases. While this strategy has proven very successful for viruses such as human immunodeficiency virus (HIV) and hepatitis C virus (HCV), the development of potent and specific antivirals requires major investments that cannot be easily duplicated to combat other existing and emerging viral pathogens. Another major challenge in antiviral drug development is the emergence of drug resistance, which can occur rapidly during monotherapy with a drug targeting a viral enzyme. This can result in dramatic failure of treatment, as seen with first generation drugs targeting the HCV protease[1]. Overall, there is a need for new antiviral strategies that circumvent these challenges by exploiting alternative mechanisms to combat viral diseases.

Targeted protein degradation has recently emerged as an alternative pharmacological strategy with several potential advantages relative to traditional occupancy-driven pharmacology[2]. Targeted degradation of a protein can be accomplished by the development of chimeric molecules commonly known as proteolysis-targeting chimeras (PROTACs) or heterobifunctional degrader molecules ("degraders"). These rationally designed small-molecules are composed of a ligand specific for the target of interest conjugated to a E3 ubiquitin ligase recruitment ligand. The degrader facilitates the heterodimerization of the two bound proteins, thus inducing the ubiquitination and subsequent proteasomal degradation of the target of interest[3]. This principle has been successfully applied to several targets, including kinases (RIPK2, BTK, BCR-ABL, CDK9)[4–7] and transcriptional enzymes (BRD4, BRD9, TRIM24)[8–10] amongst many others and is now being widely deployed as a drug discovery strategy.

Due to the analogous challenges of achieving target-specificity and circumventing resistance in both cancer and antiviral drug discovery, we postulate that targeted protein degradation might be advantageously deployed in the development of novel antivirals. As degrader molecules can operate at much lower affinities for the target than traditional occupancy-based inhibitors, they have the potential to be less susceptible to point mutations that can dramatically impact ligands requiring high target occupancy to achieve efficacy. This suggests that degrader molecules might exhibit differentiated resistance profiles relative to conventional enzymatic inhibitors[11–13]. Another potential advantage of targeted protein degradation, demonstrated for EGFR[12], is that this pharmacological mechanism abrogates all functions (enzymatic, structural, and scaffolding) of the targeted protein. This can result in higher drug potency and may be especially advantageous in addressing multifunctional viral targets. In addition, targeted protein degradation could potentially address many viral proteins with no "druggable" catalytic activity, or those whose function is unknown.

With an estimated 71 million persistently infected individuals worldwide, HCV is a main causative agent of liver diseases, including chronic hepatitis, liver cirrhosis, and hepatocellular carcinoma[14]. HCV has a single-stranded, positive-sense RNA genome, which encodes a single large polyprotein. Proteolytic cleavage of the polyprotein by cellular and viral proteases produces 10 structural-proteins and non-structural proteins. The HCV NS3 protein performs multiple essential functions during the infectious cycle[15]. The C-terminal portion of NS3 has helicase activities that are important for replication of the viral RNA genome. The N-terminal portion of NS3 together with its cofactor NS4A constitutes a serine-type protease. This viral protease cleaves the HCV polyprotein, as well as cellular proteins, such as MAVS and TRIF, two proteins involved in the induction of the

cellular immune response[16,17]. While both the protease and helicase activities of the NS3 protein are important for viral infection, only the NS3/4A protease has been successfully pursued as a drug target. Telaprevir (VX-950) is a first-generation peptidomimetic protease inhibitor, approved for HCV treatment in 2011. The α-ketoamide of telaprevir undergoes a reversible-covalent reaction with the catalytic serine to form a hemiketal[18]. The barrier to resistance of this class of protease inhibitors is low, and multiple resistant variants arose rapidly in vivo and after exposure in patients[19–21]. In some cases, resistant variants pre-exist as natural polymorphisms prior to drug treatment[22–24]. Due largely to this, telaprevir was withdrawn from the market in favor of highly effective direct-acting antivirals that can lead to complete cure after only 8 weeks of treatment[1].

To establish proof of concept for targeted protein degradation as an antiviral strategy, we selected the HCV NS3/4A protease as a well-validated HCV target. In the present work, we derive functional degraders that target the HCV NS3/4A protease using telaprevir as a protease-binding ligand. The available high-resolution structure of telaprevir with the protease (PDB ID 3SV6)[25] informed the design of suitable attachment sites for linkage to an E3-recruiting moiety. We describe here the synthesis of three degrader compounds that retain some of the parental compound's capacity to inhibit the enzymatic activity of the viral protease and additionally mediate on-target degradation of NS3. Most interestingly, the compounds exhibit antiviral activity against mutant viruses that are resistant to inhibition by the parental protease inhibitor, thus showing that this is an attractive strategy for the development of antivirals that target viral products, where barrier to resistance is a major concern.

## Results

**Design of degraders to target the HCV protease.** We synthesized bivalent small molecules based on telaprevir, a ligand for the HCV NS3/4A protease, using information gathered from the co-crystal structure of telaprevir bound to the viral protease active site[25] (Fig. 1a, b). The solvent-exposed pyrazine ring was derivatized with different linkers conjugated to chemical binders of cereblon (CRBN), the substrate receptor of the CUL4-RBX1-DDB1-CRBN E3 ubiquitin ligase complex (CRL4[CRBN]). We initially screened candidate degraders for binding to NS3 and CRBN, using inhibition of NS3/4A catalytic activity in vitro as a proxy for NS3 binding and an established cell-based assay for monitoring CRBN engagement[26] (Fig. 1), and then selected three compounds for further characterization (Fig. 1a). The CRBN-binding moieties of DGY-03-081 and DGY-04-035 were derived from lenalidomide and pomalidomide, respectively, two immunomodulatory imide drugs (IMiDs) that have been frequently deployed as CRBN binders[27]. DGY-08-097 was conjugated to a novel tricyclic imide moiety that has superior affinity for CRBN and does not induce degradation of IMiD neo-substrates such as IKZF1 and IKZF3[28]. All three compounds retain sub-micromolar $IC_{50}$ values for inhibition of HCV NS3/4A protease activity in a biochemical assay[29] (Fig. 1c). Finally, all three compounds also engage the CRL4[CRBN] complex intracellularly, as shown using a previously reported competitive displacement assay[26] (Fig. 1d) and also as evidenced by the concentration-dependent stabilization of CRBN in cells treated with DGY-08-097 in a cellular thermal shift assay (Supplementary Fig. 1).

**HCV NS3 degraders cause selective depletion of their target.** To evaluate degradation of HCV NS3 mediated by candidate small molecules in cells, we first established a cellular assay that allows detection of selective degradation of the target in the absence of other viral processes[26,30]. We constructed an inducible

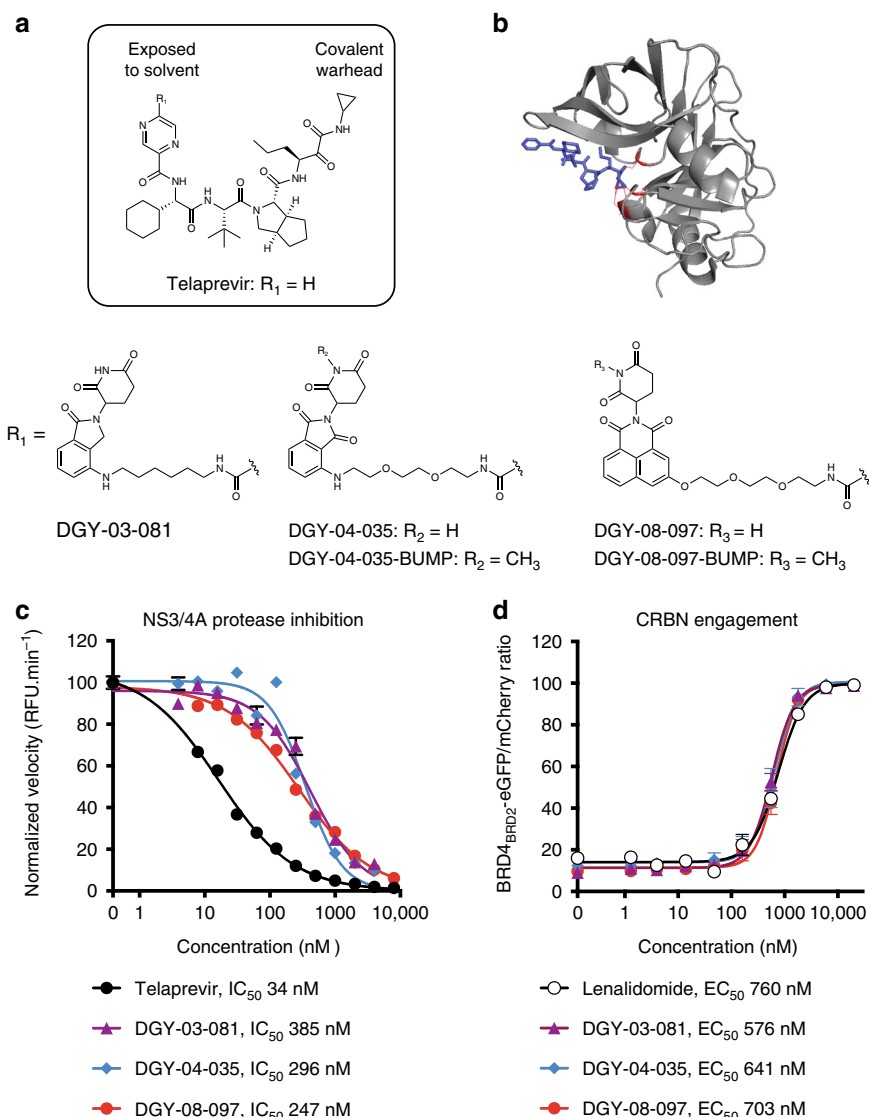

**Fig. 1** Design and biochemical characterization of the NS3-targeting degraders **a** Chemical structures of telaprevir and the degrader derivatives DGY-03-081, DGY-04-035, and DGY-08-097 where $R_1$ is the CRBN ligand. DGY-04-035-BUMP and DGY-08-097-BUMP are negative control analogs due to methylation of the glutarimide, as this modification has previously been shown to ablate CRBN-binding[11]. **b** Crystal structure of telaprevir bound to the HCV NS3/4A protease complex (PDB 3SV6). Telaprevir is shown in blue and the protease catalytic triad is shown in red. **c** Measurement of NS3/4A protease inhibition in vitro. Cell lysates containing the endogenous NS3/4A protease were prepared from cells stably expressing an HCV subgenomic replicon. These were incubated with candidate degraders at the indicated concentrations followed by addition of an HCV NS3 FRET peptide substrate. Cleavage of the FRET substrate was measured for 1 h at 30 °C, and the rate of enzymatic cleavage was fitted by linear regression. The concentration that led to a 50% decrease in HCV NS3/4A enzymatic activity ($IC_{50}$) was determined by nonlinear regression. Data are presented as means normalized to DMSO ± standard deviation of $n = 2$ technical replicates. One representative experiment is shown, with $IC_{50}$ values averaged from $n \geq 3$ independent experiments. Source data are provided as a Source Data file. RFU: relative fluorescence units. **d** Quantitative assessment of intracellular CRBN engagement using a BRD4$_{BRD2}$-eGFP-mCherry reporter assay[26]. CRBN engagement was assessed by monitoring rescue of dBET6-mediated degradation of BRD4$_{BRD2}$-eGFP. Cells stably expressing BRD4$_{BRD2}$-eGFP and mCherry were treated with 100 nM of dBET6, a specific degrader of BRD4$_{BRD2}$, and increasing concentrations of the candidate NS3 degrader. The eGFP and mCherry signals were quantified by flow cytometry analysis, and the concentration of compound that rescued 50% of BRD4$_{BRD2}$-eGFP fluorescence ($EC_{50}$) was determined by nonlinear regression. Data are presented as means normalized to DMSO ± standard deviation of $n = 3$ technical replicates. One representative experiment of $n \geq 2$ is shown

cell line expressing the full-length HCV NS3 protein fused to eGFP and linked to mCherry through the FMVD 2A ribosomal skipping sequence (Fig. 2 and Supplementary Fig. 2). In this system, measurement of the ratio of eGFP/mCherry fluorescence can be used to monitor abundance of the NS3-GFP fusion protein. We were able to demonstrate that the three candidate degraders reduce intracellular NS3 protein in a concentration-dependent manner (Fig. 2a). Consistent with previous reports[6,8],

we observed that degradation activity reached a maximum after which higher compound concentrations resulted in reduced degradation of NS3 (Fig. 2a). This "hook effect" has been attributed to independent engagement of NS3 and CRBN by the degrader, which interferes with formation of the productive trimeric complex necessary for efficient ubiquitin transfer. DGY-08-097 exhibited the most potent degradation ($DC_{50}$ of 50 nM at 4 h) of HCV NS3 in this system (Fig. 2a).

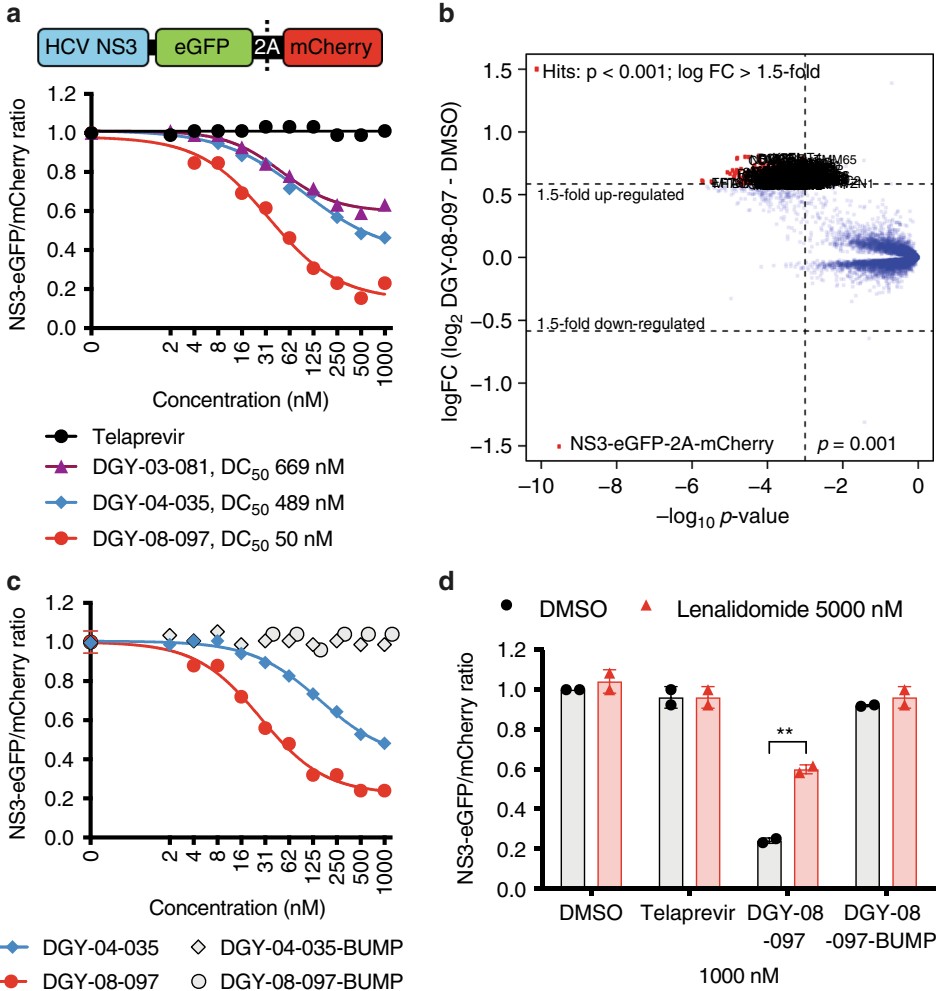

**Fig. 2** Selective degradation of HCV NS3. **a** Quantitative assessment of cellular degradation. A schematic depiction of the protein expressed in cells is shown. Expression of NS3-eGFP-2A-mCherry was induced, and cells were treated for 4 h with increasing concentrations of the indicated small molecule. eGFP and mCherry fluorescence were quantified by flow cytometry. The concentration that led to a 50% decrease in NS3-eGFP fluorescence ($DC_{50}$) was determined by nonlinear regression. One representative experiment is shown, and the $DC_{50}$ values were averaged from $n \geq 2$ independent experiments. Source data are provided as a Source Data file. **b** Quantitative proteomics analysis of cellular degradation. The scatter plot depicts the $\log_2$ fold change (FC) in protein abundance in induced cells treated for 4 h with 1000 nM DGY-08-097 compared to the DMSO control. Data shown are of a single quantitative TMT 10-plex experiment (showing ~8700 proteins, each quantified by $\geq 2$ unique peptides). Significant changes were assessed by a moderated t-test as implemented in the limma package[53]. The $\log_2$ fold change is shown on the y-axis and negative $\log_{10}$ p-values on the x-axis ($n = 3$ independent biological replicates). HCV NS3 is significantly downregulated with a $\log_2$ fold change of 1.5, and p-value of $3.08 \times 10^{-10}$. **c** Evaluation of NS3 degradation by negative control compounds that cannot engage CRBN. Expression of NS3-eGFP-2A-mCherry was induced, and cells were treated for 4 h with increasing concentrations of the indicated small molecule. eGFP and mCherry fluorescence were quantified by flow cytometry. One representative experiment is shown from $n = 2$ independent experiments. **d** Lenalidomide competition assay. Expression of NS3-eGFP-2A-mCherry was induced, and cells were treated for 4 h with 1000 nM of each telaprevir-derived compound, and co-treated with either DMSO or 5000 nM lenalidomide. The eGFP and mCherry signals were quantified by flow cytometry. Data are presented as means normalized to DMSO ± standard deviation of $n = 2$ independent experiments. Asterisks indicate that the differences between samples are statistically significant, using the unpaired t-test (**$0.001 < p < 0.01$; not significant, $p > 0.05$). Source data are provided as a Source Data file

To confirm target degradation using an orthogonal approach and to assess the selectivity of the degraders in global fashion, we performed quantitative mass-spectrometry-based proteomics following a 4 h treatment of cells with DGY-08-097 or DGY-08-097-BUMP, an analog that is unable to bind CRBN due to methylation of the glutarimide (Fig. 1a and Supplementary Fig. 3). This unbiased approach confirmed that HCV NS3 is strongly depleted in cells transiently expressing the viral protein (Fig. 2b). This effect on NS3 is specific, as no other protein of the ~8700 quantified was observed to be significantly affected by DGY-08-97; moreover, the abundance of NS3 was unaffected

by the negative control compound, DGY-08-097-BUMP (Supplementary Fig. 3). Furthermore, we detected no degradation of well-characterized IMiDs neo-substrates, such as IKZF1 and IKZF3, in the presence of DGY-08-097.

**HCV NS3 degradation requires recruitment of CRBN.** To validate the mechanism underlying HCV NS3 degradation, we interrogated the requirement for CRBN-binding using chemical perturbations. Both DGY-04-035-BUMP and DGY-08-097-BUMP inhibit the enzymatic activity of NS3/4A in vitro

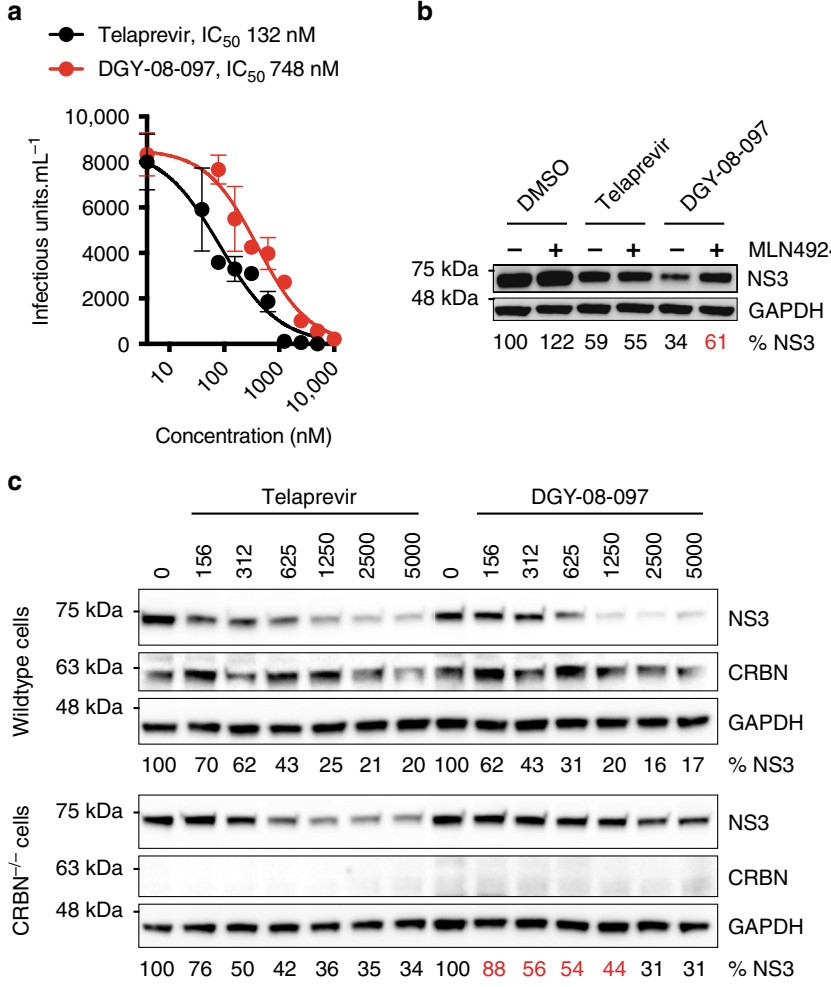

**Fig. 3** Degradation of NS3 contributes to antiviral activity. **a** NS3 degraders exhibit antiviral activity against HCV in cell culture. Huh7.5 cells were infected with HCV-JFH1-ad at a multiplicity of infection (MOI) of 0.1. The infected cells were treated from 24 to 48 h post-infection with a range of small molecule concentrations. The amount of infectious virus released to the supernatants at 48 h post-infection was measured using a 50% tissue infectious dose (TCID$_{50}$) assay. The concentration of compound that led to a 50% reduction in viral titers (IC$_{50}$) was determined by nonlinear regression. Data are presented as means ± standard error of $n = 4$ technical replicates. One representative experiment is shown, with IC$_{50}$ values averaged from $n \geq 2$ independent experiments. Source data are provided as a Source Data file. **b** Mechanistic characterization of the antiviral activity of the NS3 degraders. Huh7.5-SGR cells stably replicating the HCV JFH1 subgenomic replicon were treated with 1000 nM of the indicated small molecules for 24 h. The abundance of intracellular HCV NS3 and GAPDH was analyzed by Western blot. Source data are provided as a Source Data file. NS3 abundance was normalized to the loading control (GAPDH) and is presented as a percentage of the DMSO-treated control samples. One representative experiment is shown from $n = 2$ independent experiments. **c** Degradation of NS3 contributes to the antiviral activity of the NS3 degraders. Wildtype Huh7.5 and Huh7.5 CRBN$^{-/-}$ cells were infected with HCV-Jc1 at a MOI of 0.1. The infected cells were treated from 24 to 48 h post-infection with a range of small molecule concentrations (indicated in nM). The intracellular abundance of HCV NS3, CRBN, and GAPDH was analyzed by Western blot. Source data are provided as a Source Data file. One representative experiment is shown from $n > 3$ independent experiments. NS3 abundance was normalized to the loading control (GAPDH) and is presented as a percentage of the DMSO-treated control samples. Values represent the means of $n = 3$ independent experiments

(Supplementary Fig. 3), indicating that both compounds can engage their target, NS3; however, neither compound induced degradation of HCV NS3 in our transient assay (Fig. 2c and Supplementary Fig. 3), consistent with their loss of CRBN-binding (Supplementary Fig. 3). In addition, we found that an excess of the CRBN ligand lenalidomide rescues HCV NS3 from degradation in the presence of DGY-03-081, DGY-04-035, and DGY-08-097 (Fig. 2d and Supplementary Fig. 4). Together, these data support a mechanism in which NS3 degradation mediated by DGY-03-081, DGY-04-035, and DGY-08-097 is CRBN-dependent.

**HCV NS3 degradation contributes to antiviral activity.** Having validated the on-target and on-mechanism activities of the

telaprevir-based degraders, we next evaluated the compounds' antiviral activity in an HCV infectious assay. All three degraders exhibit antiviral activity in the absence of any cellular cytotoxicity up to a concentration of 10 to 40 µM (Fig. 3a, Supplementary Figs. 5 and 6). Antiviral activity (Fig. 3a and Supplementary Fig. 6) is correlated with potent degradation of NS3 (Fig. 2a), with DGY-08-097 exhibiting the most potent antiviral activity (IC$_{50}$ 748 nM) and the most potent degradation of NS3 while the two other degraders exhibit less potent activity in both antiviral assays (DGY-03-081 IC$_{50}$ 3069 nM; DGY-04-035 IC$_{50}$ 2920 nM) and the NS3 degradation assay (Fig. 2a).

Infectious virus production, as well as cellular accumulation of viral proteins and several other processes of the infectious cycle, are exquisitely dependent on the functions of viral enzymes.

Accordingly, treatment with telaprevir leads to reductions in intracellular viral load and intracellular NS3 protein that are solely due to inhibition of HCV NS3/4A protease activity (Fig. 3b, c). Since all three degraders were shown to enzymatically inhibit the HCV protease (Fig. 1c), we sought to parse the pharmacology derived from target degradation versus target inhibition. To ensure that the compounds were also mediating antiviral effects through their CRBN-dependent degrader functions, we assessed the requirement for an active CRL4[CRBN] E3 ubiquitin ligase complex using chemical and genetic perturbations (Fig. 3b, c). NS3 intracellular abundance was rescued by a co-treatment with the neddylation inhibitor MLN4924[31], confirming a requirement for CRL4[CRBN] activity in the degraders' antiviral effect (Fig. 3b). To further demonstrate a requirement for CRBN, we engineered a CRBN knockout Huh7.5 cell line using CRISPR editing technology. When we compared the degraders' effect on NS3 protein abundance in wildtype and CRBN$^{-/-}$ cells, we observed a significant rescue of intracellular HCV NS3 in cells deprived of CRBN, whereas telaprevir showed no dependence on CRBN for its antiviral activity (Fig. 3c and Supplementary Fig. 6). Interestingly, DGY-08-097's dose-dependent antiviral activity is observed even at concentrations equal to or greater than those at which a "hook effect" is observed in the transient NS3 degradation assay (Fig. 2a). This may be due to differences in NS3 abundance in the transient assay and the infectious virus model but may also be due to inhibition rather than degradation of the HCV NS3/4A protease at higher concentrations. These data support CRBN-dependent antiviral activity of the telaprevir-based degraders and show that we were able to derive bivalent antiviral compounds that exert their activity *via* two distinct mechanisms: inhibition and degradation of a viral enzyme.

**Degradation of NS3 variants resistant to telaprevir.** A major roadblock to the development of inhibitors that target viral products is the development of viral resistance. Numerous HCV variants have been identified that confer resistance to telaprevir and other ketoamide compounds, and the basis for this resistance has been thoroughly characterized[22]. A mutation at residue A156 was notably shown to extend the bond between the α-ketoamide warhead and the catalytic serine, thus potentially reducing the capacity for covalent modification and thus decreasing inhibitor potency[25]. The hydrophobic side chains of this class of inhibitors fill several substrate binding pockets, and both high resolution structures[25,32] and molecular dynamics simulations[33,34] suggest that the complementarity of these interactions may be affected by resistance mutations at the underlying residues V36, T54, and V55. To evaluate whether targeted protein degradation can confer antiviral activity against telaprevir-resistant viruses, we evaluated the activity of the telaprevir-based degraders against HCV clones bearing the V55A or A156S mutations (HCV-NS3-V55A and HCV-NS3-A156S, respectively). Interestingly, while both mutants exhibit resistance to telaprevir treatment, we observed that degrader DGY-08-097 retains an antiviral effect against both mutant viruses and reduces NS3 abundance and infectious virus production (Fig. 4). We note that while the NS3-A156S mutation reduces sensitivity to DGY-08-097, the change in antiviral potency is only 3-fold (wildtype, IC$_{50}$ 558 nM; NS3-A156S, IC$_{50}$ 1561 nM), while it is 10-fold for telaprevir (wildtype, IC$_{50}$ 98 nM; NS3-A156S, IC$_{50}$ 949 nM). The NS3-V55A mutant is known to exist as a natural polymorphism even in the absence of drug treatment[22,23]. Consistent with prior reports, we observed that this variant confers modest resistance to telaprevir, with only a 3-fold change in antiviral potency (wildtype IC$_{50}$ 98 nM; NS3-V55A IC$_{50}$ 288 nM). Remarkably, DGY-08-097 has

similar antiviral activities against the wildtype virus and the NS3-V55A mutant (wildtype, IC$_{50}$ 558 nM; NS3-V55A, IC$_{50}$ 508 nM). Together, these results provide proof-of-concept for the superiority of small molecule degraders in mediating antiviral activity against drug-resistant viruses that arise in the presence of conventional direct-acting antivirals that target enzymatic activity.

## Discussion

Targeted protein degradation has emerged as a powerful new drug discovery strategy. This approach involves developing small molecules that can catalyze the ubiquitin-mediated degradation of target proteins of interest. These small molecules have either been identified serendipitously, such as thalidomide[35] or indisulam[36,37], or designed intentionally through the development of bivalent molecules that tether a known ligand of the target with an E3-complex-recruiting moiety. In this study we took this second approach to develop antivirals that act by inducing degradation of an antiviral target, the HCV NS3/4A viral protease. In this work, we focused our effort on development of degraders of a virus-encoded (rather than host-encoded) protein because the vast majority of successful antiviral drugs today are direct-acting antivirals, including those that now cure HCV and enable long-term control of HIV. Of these, the most frequent targets of direct-acting antivirals have been viral polymerases and proteases due to their well characterized enzymatic activity and decades of drug discovery experience building ligands for these target classes. Targeted protein degradation has a number of potential advantages that make the approach complementary to traditional catalytic inhibitors. First, targeted protein degraders have been shown to operate at sub-stoichiometric doses due to their event-driven pharamacology[4]. They thus might potentially be less sensitive to point mutations that lead to significant losses of affinity and hence potency for traditional occupancy-dependent inhibitors. Second, similar to genetic deletion approaches, targeted degraders ablate all protein function including non-enzymatic functions, which are frequently found in virally encoded proteins.

Here, we have synthesized and characterized bifunctional degraders of the HCV NS3 protein. We developed a chemical strategy focused on the mechanism of degradation of virally expressed NS3 by exploiting the ubiquitin ligase activity of the CRL4[CRBN] complex. We have shown that our optimal bivalent degrader, DGY-08-097, induces rapid and sustained proteasome-mediated degradation of NS3 that correlates with inhibition of infectious HCV in a cellular infection model. DGY-08-097 is selective for NS3 as assessed by quantitative expression proteomics. We demonstrated that degradation of NS3 is a key contributor to the pharmacology of DGY-08-097 as assessed using control compounds unable to bind CRBN, competition with CRBN ligands, and CRBN-deficient cells. Most excitingly, we demonstrate that DGY-08-097, a first-generation chemical probe, retains antiviral activity against NS3 point mutations that confer resistance to the highly optimized protease inhibitor drug, telaprevir. This confirms that these small molecule degraders are less susceptible to mutations that impact ligand binding and could thus be deployed to suppress or to treat viral variants that are associated to resistance to conventional inhibitors.

This proof-of-concept study suggests that targeted protein degradation is an approach worthy of further exploration as an anti-viral strategy. Interestingly, although DGY-08-097 exhibits a "hook effect" with decreased activity at higher concentrations in the transient NS3 degradation assay (Fig. 2a), we observe no such effect in the antiviral assays. While this is presumably due to differences in target abundance and hence tertiary complex

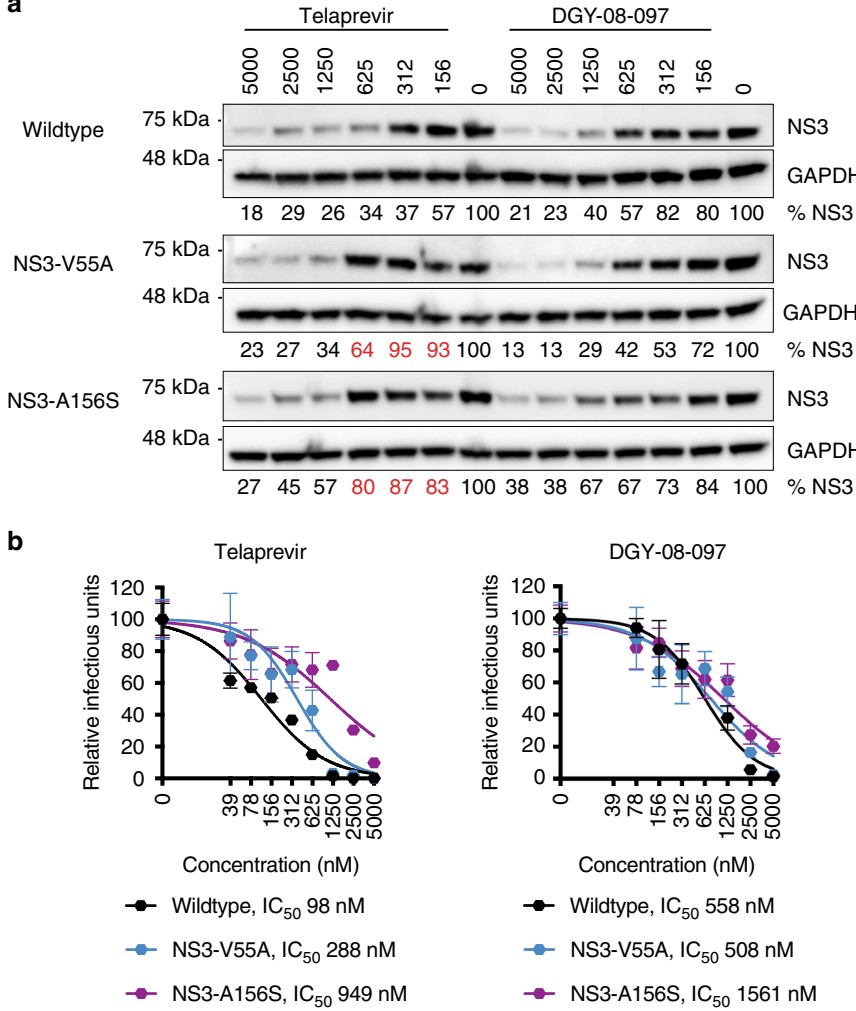

**Fig. 4** The NS3 degraders inhibit telaprevir-resistant HCV. Huh7.5 cells were infected with wildtype HCV-Jc1 or the indicated telaprevir-resistant viruses at a MOI of 0.1. The infected cells were treated from 24 to 48 h post-infection with a range of small molecule concentrations (indicated in nM). **a** HCV NS3 and GAPDH abundance was evaluated by Western blot. Source data are provided as a Source Data file. One representative experiment is shown from $n = 4$. NS3 abundance was normalized to the loading control (GAPDH) and is presented as a percentage of the DMSO-treated control samples. Values represent the means of $n = 4$ independent experiments. **b** The amount of infectious virus released to the supernatants at 48 h post-infection was measured using a 50% tissue infectious dose ($TCID_{50}$) assay. The concentration of compound that led to a 50% reduction in viral titers ($IC_{50}$) was determined by nonlinear regression. Data are presented as means normalized to DMSO ± standard error of $n = 4$ technical replicates. One representative experiment is shown, with $IC_{50}$ values averaged from $n \geq 2$ independent experiments

formation in the transient and infectious virus assays or to inhibition of HCV NS3/4A protease activity at higher concentrations, other small molecule degraders have demonstrated efficacy in preclinical in vivo models of cancer despite having well-documented "hook effects" in vitro[7]. Further work is needed to define therapeutic windows for antiviral activity in vivo. While using known ligands such as telaprevir provides a convenient entry point, exciting potential also exists in developing degraders that exploit non-catalytic binding pockets. This would enable degradation of targets that do not have a conventionally druggable pocket and would enable combination of degraders with traditional inhibitors to develop durable antiviral drug regimens. In particular, many viruses express essential proteins for which structural data are insufficient to enable conventional structure-based inhibitor design and our knowledge of biochemical function is too limited to allow development of target-based screening assays. The targeted protein degradation strategy may make these "undruggable" proteins tractable targets by allowing discovery of ligands that can be converted to degraders.

## Methods

**Cells**. All cells were cultured in a 37 °C incubator with 5% $CO_2$ and routinely examined to be free of mycoplasma contamination. Huh7.5 cells[38] were obtained from Charles Rice (Rockefeller University) and cultured in Dulbecco's Modified Eagle's medium (DMEM) supplemented with nonessential amino acids and 10% fetal bovine serum (FBS). Huh7.5-SGR cells stably replicating the HCV JFH1 subgenomic replicon were produced as previously described[39] and cultured in DMEM supplemented with nonessential amino acids, 10% FBS and 0.9 mg.mL$^{-1}$ G418 (Teknova G5005). Flp-In$^{TM}$ 293 cells (Thermo Fisher Scientific R75007) stably expressing the BRD4$_{BD2}$-GFP with mCherry reporter[26] and HEK293T cells (ATCC, CRL-3216) were cultured in DMEM supplemented with 10% FBS.

The CRISPR-mediated CRBN knockout Huh7.5 cell line was generated using gene-editing vectors kindly provided by James Bradner. Huh7.5 cells were transfected with the PX458 plasmid[40] containing the sgCRBN-496 guide[41] using Lipofectamine 2000 (Thermo Fisher Scientific 11668019). GFP$^+$ populations were enriched by flow cytometry, and CRBN-knockout was validated by Western blotting.

Flp-In$^{TM}$ T-REx$^{TM}$ 293 cells (Thermo Fisher Scientific R78007) were cultured in DMEM supplemented with 10% FBS, 100 μg/mL Zeocin$^{TM}$ (Thermo Fisher Scientific R25001) and 15 μg/mL blasticidin (Invivogen ant-bl-1). Plasmid pcDNA5/FRT/TO-JFH1-NS3-eGFP-2A-mCherry was constructed as follow. A PCR fragment was amplified from pJFH1[42] using primers NotI-HCV-3431-FW and KpnI-HCV-5323-RV (primers sequences are provided in the

Supplementary Methods), digested with NotI and KpnI, and cloned into a similarly treated pcDNA5/FRT-MCS-eGFP-P2A-mCherry[26] plasmid to yield pcDNA5/FRT-JFH1-NS3-eGFP-2A-mCherry. We next constructed pcDNA5/FRT/TO-JFH1-NS3-eGFP-2A-mCherry using NEBuilder High Fidelity DNA Assembly Master Mix (New England Biolabs E2621) and PCR fragments amplified from pcDNA$^{TM}$5/FRT/TO and pcDNA5/FRT-JFH1-NS3-eGFP-2A-mCherry using primer pairs pcDNA-FRT-TO-GA-FW/pcDNA-FRT-TO-GA-RV and HCV-NS3-GA-FW / HCV-NS3-GA-RV, respectively (primers sequences are provided in the Supplementary Methods). Stable cell lines expressing the HCV NS3-eGFP protein fusion and the mCherry reporter were generated using the Flp-In$^{TM}$ System. Flp-In$^{TM}$ T-REx$^{TM}$ 293 cells were transfected with pcDNA5/FRT/TO-JFH1-NS3-eGFP-2A-mCherry and pOG44 (Thermo Fisher Scientific V600520) at a ratio 1:9, using Lipofectamine 2000 (Thermo Fisher Scientific 11668019). Cells were selected with 50 µg.mL$^{-1}$ hygromycin B (MilliporeSigma H3274) and 15 µg.mL$^{-1}$ blasticidin (Invivogen ant-bl-1). After expansion, expression of the fusion protein was induced for 24 h using 10 µg.mL$^{-1}$ doxycycline (MilliporeSigma D9891) and GFP+/mCherry+ cells were single-cell sorted by flow cytometry into 96-well plates. After 2–3 weeks, cell clones that grew up from each individual well were expanded and validated by Western blotting analysis.

**Production of recombinant viruses**. The infectious cDNA clone of HCV strain JFH1 was kindly provided by Takaji Wakita[42]. The cell culture adaptive mutation V2440L[43] was introduced through site-directed mutagenesis using primers HCV (V2440L)-7643-FW and HCV(V2440L)-7673-RV (primers sequences are provided in the Supplementary Methods), yielding plasmid pJFH1-ad. The infectious cDNA clone of the HCV Jc1 chimera, pJc1, was kindly provided by Brett Lindenbach[44]. We introduced the NS3-V55A and NS3-A156S mutations through site-directed mutagenesis of pJc1, using, respectively, the primer pairs HCV(V55A)-3578-FW / HCV(V55A)-3606-RV and HCV(A156S)-3880-FW / HCV(A156S)-3911-RV (primers sequences are provided in the Supplementary Methods). In vitro transcripts were synthesized from XbaI-linearized, mung bean nuclease-treated pJFH1-ad, pJc1, pJc1(NS3-V55A) or pJc1(NS3-A156S) plasmids using an AmpliScribe T7 Flash Transcription kit (Epicentre ASF3257).

For virus production, Huh7.5 cells were washed twice in phosphate-buffered saline (PBS), and $2 \times 10^6$ cells were electroporated with 2 µg HCV in vitro transcripts in 4 mm gap cuvettes using an ECM 830 electroporator (BTX Harvard Apparatus) at the following settings: 5 pulses at 820 V, 100 µs per pulse with 1.1 s intervals. Following electroporation, the cells were seeded in a T25 flask in DMEM supplemented with nonessential amino acids and 10% FBS. Supernatants containing infectious virus were recovered on day 3 post-electroporation (passage 0 viral stock). Electroporated cells were next washed, lifted with trypsin, and transferred to a T75 flask. Supernatants containing infectious virus were recovered on day 3 post-splitting (passage 1 viral stock). All viral stocks were clarified by centrifugation at $1000 \times g$ for 5 min, and aliquots were stored at −80 °C. All stocks used for experiments were at passage 0 or 1. All work with infectious virus was performed in a biosafety level 2 (BSL2) laboratory using additional safety practices as approved by the Harvard Committee on Microbiological Safety.

The presence of the NS3-V55A and NS3-A156S mutations in the viral stocks was verified as follows. Viral RNA was extracted from supernatants using a QIAamp viral RNA extraction kit (Qiagen 52906). Part of the HCV genome was amplified using the SuperScript® III One-Step RT-PCR System with Platinum® Taq DNA Polymerase (Thermo Fisher Scientific 12574–018) using the primers HCV-3431-FW and HCV-5323-RV (primers sequences are provided in the Supplementary Methods). The PCR fragment was then sequenced at the DF/HCC DNA sequencing facility.

Both mutant viruses exhibited resistance to telaprevir, and the fold-changes in antiviral potency were in line with previously reported values[34,45,46].

**Compound synthesis and characterization**. Dimethyl sulfoxide (DMSO), telaprevir, lenalidomide, pomalidomide and MLN4924 were, respectively, purchased from MilliporeSigma (D8418), MedChemExpress (HY-10235), Ark Pharm (AK-47482 and AK-104087) and Thermo Fisher Scientific (5054770001). dBET6 and 3-110-22 were synthesized using previously reported methods[8,47].

Detailed descriptions of synthetic methods and compound characterization are provided in the Supplementary Methods.

Compounds were evaluated for their pan-assay interference activity by counter-screening against the malate dehydrogenase enzyme (MDH) a frequently used assessment of compounds that exhibit Pan-Assay interference (PAINs)[48]. Briefly, small molecules were serially diluted (2-fold dilution series from 100 µM) and were mixed with 200 µM oxaloacetic acid and 200 µM NADH in working buffer (100 mM potassium phosphate, pH 7.4). The final concentration of DMSO was 2% for all samples. The reaction was initiated by adding 0.6 nM MDH (EMD Millipore 442610) and absorbance was immediately monitored at 340 nm for 5 min using a SpectraMax Plus 384 microplate reader (Molecular Devices). The enzymatic velocities were determined by linear regression analysis (GraphPad Software). None of the compounds showed any inhibitory activity against MDH (Supplementary Fig. 7). 3-110-22, a compound previously shown to exhibit non-specific MDH inhibition[49], was used as a positive control.

**Antibodies**. Mouse monoclonal antibody 9E10 anti-HCV NS5A was kindly provided by Charles Rice[38] and diluted 1:1000 in TCID$_{50}$ assays. Commercial antibodies used in Western blotting, along with their respective dilutions, are listed as follow. Mouse monoclonal antibody anti-HCV NS3 was purchased from Abcam (ab65407, 1:3,000). Mouse monoclonal antibody against GAPDH, rabbit polyclonal antibody against CRBN, and rabbit polyclonal antibody against SOD1 were respectively purchased from GeneTex (GTX28245, 1:5,000), Novus Biologicals (NBP1-91810, 1:500), and Sigma-Aldrich (HPA00140-1; 1:500). Horseradish peroxidase (HRP)-conjugated goat anti-mouse IgG and anti-rabbit IgG antibodies were obtained from Bio-Rad Laboratories (170-6516 and 170-6515, respectively, 1:3,000). IRDye® 800CW-conjugated goat anti-rabbit antibody was purchased from LI-COR (926-32211, 1:10,000).

**HCV NS3/4A protease enzymatic assay**. Enzymatic inhibition experiments were adapted from previously described methods[29]. Huh7.5-SGR cells were seeded at a density of $1 \times 10^4$ cells/well in a white 96-well plate. Test compounds were serially diluted in lysis buffer (1× Luciferase Cell Culture Lysis Reagent (Promega E1531) supplemented with 0.15 M NaCl). The Huh7.5-SGR cells were washed once with PBS and lysed with 90 µL of lysis buffer supplemented with small molecules for 15 min at room temperature. Protease reactions were initiated by adding 1 µM HCV NS3/4A substrate [Ac-DE-D(Edans)-EE-Abu-ψ-[COO]AS-K(Dabcyl)-NH$_2$] (AnaSpec AS-22991) and monitored for 60 min using a Synergy plate reader (BioTek) at excitation and emission wavelengths of 355 nm and 495 nm, respectively. The initial cleavage velocities were determined by linear regression analysis (GraphPad Software). Concentrations resulting in 50% inhibition (IC$_{50}$) were calculated using the nonlinear fit variable slope model (GraphPad Software).

**Competitive displacement assay for cellular CRBN engagement**. Cells stably expressing the BRD4$_{BD2}$-GFP with mCherry reporter[26] were seeded at 30-50% confluency in 384-well plates with 50 µL FluoroBrite DMEM media (Thermo Fisher Scientific A18967) containing 10% FBS per well a day before compound treatment. Compounds and 100 nM dBET6 were dispensed using a D300e Digital Dispenser (HP), normalized to 0.5% DMSO, and incubated with cells for 5 h. The assay plate was imaged immediately using an Acumen High Content Imager (TTP Labtech) with 488 nm and 561 nm lasers in $2 \times 1$ µm grid per well format. The resulting images were analyzed using CellProfiler[50]. A series of image analysis steps ('image analysis pipeline') was constructed. First, the red and green channels were aligned and cropped to target the middle of each well (to avoid analysis of heavily clumped cells at the edges), and a background illumination function was calculated for both red and green channels of each well individually and subtracted to correct for illumination variations across the 384-well plate from various sources of error. An additional step was then applied to the green channel to suppress the analysis of large auto fluorescent artifacts and enhance the analysis of cell specific fluorescence by way of selecting for objects under a given size, 30 A.U., and with a given shape, speckles. mCherry-positive cells were then identified in the red channel filtering for objects between 8–60 pixels in diameter and using intensity to distinguish between clumped objects. The green channel was then segmented into GFP positive and negative areas and objects were labeled as GFP positive if at least 40% of it overlapped with a GFP positive area. The fraction of GFP-positive cells/mCherry-positive cells in each well was then calculated, and the green and red images were rescaled for visualization. The values for the concentrations that lead to a 50% increase in BRD4$_{BD2}$-eGFP accumulation (EC$_{50}$) were calculated using the nonlinear fit variable slope model (GraphPad Software).

**Cellular thermal shift assay for CRBN engagement**. HEK293T cells were trypsinized, washed with PBS, and suspended in PBS supplemented with cOmplete protease inhibitor cocktail (MilliporeSigma 11697498001)

For the thermal shift assay, suspended cells ($2 \times 10^4$ cells/µL) were treated with 10 µM lenalidomide, DGY-08-097 or DMSO control for 1 h at 37 °C with shaking at 200 rpm. Each of the three treatments was then separated into 8 fractions of 100 µL, each containing $2 \times 10^6$ cells for thermal proteome profiling. Fractions were heated at the indicated temperatures (46–70 °C) for 3 min, followed by 3 min incubation at room temperature and snap freezing in liquid nitrogen.

For the dose-response assay, suspended cells were split into 11 fractions of $2 \times 10^6$ cells each and treated with the half-maximal dose of DGY-08-097 (1 nM–50 µM) for 1 h at 37 °C with shaking at 200 rpm. Fractions were heated at 61 °C for 3 min, followed by 3 min incubation at room temperature and snap freezing in liquid nitrogen.

Samples were lysed with five freeze thaw cycles using dry ice and 30 °C dry bath. Samples were centrifuged in a mini centrifuge for 30 min at 4 °C to separate the aggregated proteins from the soluble protein fraction. Supernatants were collected and equal amounts were used for Western blotting analysis using LI-COR reagents.

**CRBN-binding assay**. Measurement of compounds binding to CRBN was performed by fluorescence polarization[26]. Atto565-conjugated lenalidomide (10 nM) was mixed with 100 nM of purified DDB1ΔB-CRBN in 50 mM Tris pH 7.5, 200 mM NaCl, 0.1% Pluronic F-68 solution, 1 mM TCEP, in 384-well microplates. Compounds were dispensed using a D300e Digital Dispenser (HP), normalized to 1% DMSO, and incubated for 60 min at room temperature. The change in

fluorescence polarization was monitored using a PHERAstar FS microplate reader (BMG Labtech) for 20 min in 200 s cycles. The values for the concentrations that lead to a 50% inhibition ($IC_{50}$) were calculated using the nonlinear fit variable slope model (GraphPad Software).

**Cellular NS3 degradation assay**. Stable cells expressing the HCV NS3-eGFP protein fusion and the mCherry reporter were seeded at a density of $1 \times 10^5$ cells/ well in a 24-well plate. Expression of the fusion protein was induced for 24 h using 1 µg.mL$^{-1}$ tetracycline (MilliporeSigma T3383). Cells were incubated with the indicated compounds for 4 h. For flow cytometry analysis, cells were lifted with Versene (Thermo Fisher Scientific 15040-066) and resuspended in PBS. The cells were analyzed using a BD$^{TM}$ LSR II flow cytometer (BD Biosciences). Signal from at least 10,000 events per sample was acquired, and the eGFP and mCherry fluorescence monitored. Data were analyzed using FlowJo (FlowJo, LCC). Forward and side scatter outliers, frequently associated with cell debris, were removed leaving >90% of total cells, which was followed by removal of eGFP and mCherry signal outliers, leaving 55–85% of total cells as the set used for quantification. The eGFP protein abundance relative to mCherry was quantified as a ten-fold amplified ratio for each individual cell using the formula: $10 \times$ eGFP/mCherry. The median of the ratio was then calculated per set, normalized to the median of the DMSO ratio. The values for the concentrations resulting in 50% degradation ($DC_{50}$) were calculated using the nonlinear fit variable slope model (GraphPad Software). An unpaired $t$-test was used to compare quantitative data (GraphPad Software). Statistically significant differences between experimental samples and DMSO-treated samples are shown by asterisks in the figures (***$p > 0.001$; **$0.001 < p < 0.01$; not significant, $p > 0.05$).

**Sample preparation TMT LC-MS3 mass spectrometry**. Stable cells expressing the HCV NS3-eGFP protein fusion and the mCherry reporter were seeded at a density of $5 \times 10^6$ cells/well in a T75 flask. Expression of the fusion protein was induced for 24 h using 1 µg.mL$^{-1}$ tetracycline (MilliporeSigma T3383). Cells were treated in biological triplicates for 4 h with DMSO, DGY-08-097 (1 µM) or DGY-08-097-BUMP (1 µM). The cells were lifted with Versene (Thermo Fisher Scientific 15040-066), washed with PBS twice, and pellets were snap-frozen at −80 °C until processed.

Lysis buffer (8 M urea, 50 mM NaCl, 50 mM 4-(2-hydroxyethyl)-1-piperazineethanesulfonic acid (EPPS) pH 8.5, supplemented with protease and phosphatase inhibitors (Roche)) was added to the cell pellets, which were then homogenized by 20 passes through a 21 gauge (1.25 mm long) needle to achieve a cell lysate with a protein concentration between 1–4 mg.mL$^{-1}$. A micro-BCA assay (Pierce) was used to determine the final protein concentration in the cell lysate. 200 µg of protein for each sample was reduced and alkylated as previously described[51].

Proteins were precipitated using methanol/chloroform. In brief, four volumes of methanol were added to the cell lysate, followed by one volume of chloroform, and finally three volumes of water. The mixture was vortexed and centrifuged to separate the chloroform phase from the aqueous phase. The precipitated protein was washed with three volumes of methanol, centrifuged, and the resulting washed precipitated protein was allowed to air dry. Precipitated protein was resuspended in 4 M urea, 50 mM HEPES pH 7.4, followed by dilution to 1 M urea with the addition of 200 mM EPPS, pH 8. Proteins were first digested with LysC (1:50; enzyme:protein) for 12 h at room temperature. The LysC digestion was diluted to 0.5 M urea with 200 mM EPPS pH 8 followed by digestion with trypsin (1:50; enzyme:protein) for 6 h at 37 °C. Tandem mass tag (TMT) reagents (Thermo Fisher Scientific) were dissolved in anhydrous acetonitrile (ACN) according to manufacturer's instructions. Anhydrous ACN was added to each peptide sample to a final concentration of 30% v/v, and labeling was induced with the addition of TMT reagent to each sample at a ratio of 1:4 peptide:TMT label. The 10-plex labeling reactions were performed for 1.5 h at room temperature and the reaction quenched by the addition of hydroxylamine to a final concentration of 0.3% for 15 min at room temperature. The sample channels were combined at a 1:1:1:1:1:1:1:1:1:1 ratio, desalted using C$_{18}$ solid phase extraction cartridges (Waters) and analyzed by LC-MS for channel ratio comparison. Samples were then combined using the adjusted volumes determined in the channel ratio analysis and dried down in a speed vacuum. The combined sample was then resuspended in 1% formic acid, and acidified (pH 2−3) before being subjected to desalting with C18 SPE (Sep-Pak, Waters). Samples were then offline fractionated into 96 fractions by high pH reverse-phase HPLC (Agilent LC1260) through an aeris peptide xb-c18 column (Phenomenex) with mobile phase A containing 5% acetonitrile and 10 mM NH$_4$HCO$_3$ in LC-MS grade H$_2$O, and mobile phase B containing 90% acetonitrile and 10 mM NH$_4$HCO$_3$ in LC-MS grade H$_2$O (both pH 8.0). The 96 resulting fractions were then pooled in a non-continuous manner into 24 fractions, and these fractions were used for subsequent mass spectrometry analysis.

Data were collected using an Orbitrap Fusion Lumos mass spectrometer (Thermo Fisher Scientific, San Jose, CA, USA) coupled with a Proxeon EASY-nLC 1200 LC pump (Thermo Fisher Scientific). Peptides were separated on an EasySpray ES803 75 µm inner diameter microcapillary column (Thermo Fisher Scientific). Peptides were separated using a 190 min gradient of 6–27% acetonitrile in 1.0% formic acid with a flow rate of 350 nL.min$^{-1}$.

Each analysis used a MS3-based TMT method[52]. The data were acquired using a mass range of $m/z$ 340–1350, resolution 120,000, AGC target $5 \times 10^5$, maximum injection time 100 ms, dynamic exclusion of 120 s for the peptide measurements in the Orbitrap. Data-dependent MS2 spectra were acquired in the ion trap with a

normalized collision energy (NCE) set at 55%, AGC target set to $1.5 \times 10^5$ and a maximum injection time of 150 ms. MS3 scans were acquired in the Orbitrap with a HCD collision energy set to 55%, AGC target set to $1.5 \times 10^5$, maximum injection time of 150 ms, resolution at 50,000, and with a maximum synchronous precursor selection (SPS) precursors set to 10.

**LC-MS data analysis**. Proteome Discoverer 2.2 (Thermo Fisher Scientific) was used for .RAW file processing and controlling peptide and protein level false discovery rates, assembling proteins from peptides, and protein quantification from peptides. MS/MS spectra were searched against a Uniprot human database (September 2016) containing the HCV NS3 protein sequence with both the forward and reverse sequences. Database search criteria are as follows: tryptic with two missed cleavages, a precursor mass tolerance of 20 ppm, fragment ion mass tolerance of 0.6 Da, static alkylation of cysteine (57.02146 Da), static TMT labeling of lysine residues and N-termini of peptides (229.16293 Da), and variable oxidation of methionine (15.99491 Da). TMT reporter ion intensities were measured using a 0.003 Da window around the theoretical $m/z$ for each reporter ion in the MS3 scan. Peptide spectral matches with poor quality MS3 spectra were excluded from quantitation (summed signal-to-noise across 10 channels <200 and precursor isolation specificity <0.5), and resulting data was filtered to only include proteins that had a minimum of 2 unique peptides identified. Reporter ion intensities were normalized and scaled using in-house scripts in the R framework[53]. Statistical analysis was carried out using the limma package within the R framework[53].

**Viral infection**. All work with infectious virus was performed in a biosafety level 2 (BSL2) laboratory using additional safety practices as approved by the Harvard Committee on Microbiological Safety. Huh7.5 or Huh7.5 CRBN$^{-/-}$ cells were seeded at a density of $25 \times 10^3$ cells/well in a 48-well plate. The cells were infected at MOI of 0.1 by incubation at 37 °C with viral inoculum diluted in DMEM supplemented with nonessential amino acids and 10% FBS. At 24 h post-infection, the inoculum was removed, and complete medium containing the indicated concentrations of small molecules was then overlaid on cells. Cell lysates and viral supernatants were harvested at 48 h post-infection.

**Tissue culture infectious dose assay (TCID$_{50}$)**. The yield of infectious particles was quantified by measuring the 50% tissue culture infective dose (TCID$_{50}$)[54]. Briefly, Huh7.5 cells were seeded at a density of $5 \times 10^3$ cells/well in 96-well plates. Aliquots from HCV infections were serially diluted in complete medium, and 150 µL of each dilution was added to the cells, followed by incubation at 37 °C for 4 days. The medium was aspirated, and the cells were washed with PBS and fixed with methanol for 15 min at −20 °C. After fixation, the cells were washed with PBS and incubated overnight at 4 °C with monoclonal antibody 9E10 against HCV NS5A. The cells were then washed and incubated with HRP-conjugated anti-mouse IgG antibody. The plates were developed with the Vector VIP peroxidase substrate kit (Vector Laboratories SK-4600). The number of infectious units per mL was evaluated by counting the number of foci under a microscope. The values for the concentrations that lead to 50% inhibition ($IC_{50}$) were calculated using the nonlinear fit variable slope model (GraphPad Software).

**Western blotting**. Protein lysates were prepared by cell lysis in radio-immunoprecipitation assay buffer (Boston BioProducts BP-419) containing protease inhibitors (MilliporeSigma 6538282001). Equal amounts of proteins were fractionated on a 4–20% polyacrylamide gel (GenScript) and transferred to a polyvinylidene difluoride membrane (MilliporeSigma IPVH00010) using the Trans-Blot Turbo Transfer System (Bio-Rad Laboratories). For the cellular thermal shift experiments, equal amounts of samples were fractionated on an AnyKD mini-PROTEAN TGX Precast Protein Gel (Bio-Rad Laboratories) and transferred to a polyvinylidene difluoride membrane (Life Technologies, IB24001) using the iBlot 2.0 dry blotting system (Thermo Fisher Scientific).

When using HRP-conjugated antibodies, the membrane was blocked for 1 h at room temperature in PBS-Tween 20 (PBS-T) plus 5% (wt/vol) milk, then incubated overnight at 4 °C with appropriate dilutions of the primary antibodies. The membrane was then washed in PBS-T, followed by incubation for 1 h at room temperature in the presence of HRP-conjugated secondary antibodies. After washes in PBS-T, the membrane was developed with enhanced chemiluminescence reagents (Thermo Fisher Scientific PI32106), and the signal was captured using the Amersham Imager 600 (GE Healthcare).

When using LI-COR reagents, membranes were blocked with LI-COR blocking solution (LI-COR), and incubated with primary antibodies overnight, followed by three washes in LI-COR blocking solution and incubation with secondary antibodies for 1 h in the dark. After three final washes, the membranes were imaged on a LI-COR fluorescent imaging station (LI-COR).

Band intensities were quantified using ImageJ software (http://imagej.nih.gov/ij/). An unpaired t-test was used to compare quantitative data (GraphPad Software). Statistically significant differences between experimental samples and DMSO-treated samples are shown by asterisks in the figures (***$p > 0.001$; **$0.001 < p < 0.01$; not significant, $p > 0.05$).

**Cytotoxicity assay**. Huh7.5-SGR cells were seeded at a density of $1 \times 10^4$ cells/well in a white 96-well plate. Small molecules were serially diluted in DMEM supplemented with 2% FBS. The mixture was added to the cells, and the plates were incubated at 37 °C for 24 h. The cytotoxicity was measured by quantitation of the ATP present, using a CellTiter-Glo® luminescent cell viability assay (Promega G7572). The values for the concentrations that lead to 50% cytotoxicity ($CC_{50}$) were calculated using the nonlinear fit variable slope model (GraphPad Software).

**Reporting summary**. Further information on research design is available in the Nature Research Reporting Summary linked to this article.

## Data availability

All data supporting the findings of this study are available from the corresponding author upon reasonable request. Mass spectrometry raw data files have been deposited in the PRIDE Archive under the data set identifier PXD014346. The source data underlying Figs. 1c, 2a, d, 3a-c, 4a, b and Supplementary Figs. 1a, b, 2c, 3d, 4a-c, 5, 6a-c, are provided as a Source Data file.

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

## Acknowledgements

This work was supported by an award from the Quadrangle Fund for the Advancement and Seeding of Translational Research at Harvard Medical School (Q-FASTR) to P.L.Y., the Dana-Farber Cancer Institute Medicinal Chemistry Core and NIH award U19AI109740 to N.S.G, and NIH NCI R01 CA214608-02 to E.S.F. E.S.F. is a Damon Runyon-Rachleff Innovator supported in part by the Damon Runyon Cancer Research Foundation (DRR-50-18). The authors thank Charles Rice (Rockefeller University) and Takaji Wakita (Tokyo Metropolitan Institute of Neuroscience) for access to HCV JFH1 replicons and infectious cDNA clone, 9E10 antibody and Huh7.5 cells; Brett Lindenbach for access to the HCV Jc1 clone; Feng Zhang for the pSpCas9(BB)-2A-GFP (PX458) plasmid (Addgene plasmid # 48138); James Bradner for plasmids to produce the CRBN$^{-/-}$ Huh7.5 cell line.

## Author contributions

T.Z., N.S.G., and P.L.Y. conceived the project. M.d.W., G.D., K.A.D., T.Z., E.S.F., N.S.G., and P.L.Y. designed experiments. G.D. performed chemical synthesis. M.d.W., K.A.D., N. A.E., J.C.Y., J.K., and R.P.N. performed biological evaluation experiments and analyzed data. K.A.D performed the thermal proteome profiling experiments. K.A.D. and N.A.E performed MS proteomics experiments. M.d.W., N.S.G., and P.L.Y. wrote the paper with input from all authors. All authors reviewed and edited the paper.

## Additional information

**Competing interests:** N.S.G. is an equity holder and scientific advisor for Syros, Gatekeeper, Soltego, C4, B2S, Petra and Aduro companies. N.S.G., G.D., P.L.Y., M.d.W. are inventors on a patent covering the compounds described in this paper. E.S.F is an equity holder and scientific advisor for C4 Therapeutics and Civetta Therapeutics and is a consultant to Novartis, AbbVie, Pfizer, and Deerfield. E.S.F received research funding from Novartis and Astellas not related to this work. The remaining authors declare no competing interests.

