## [Peer Review File · Nature Communications]

Editorial Note: This manuscript has been previously reviewed at another journal that is not operating a transparent peer review scheme. This document only contains reviewer comments and rebuttal letters for versions considered at Nature Communications .

Reviewers' Comments:

Reviewer #4:

Remarks to the Author:

The reviewers have satisfied the comments made by Reviewer #2

Reviewer #5:

Remarks to the Author:

This manuscript focuses on an emerging drug approach that relies on targeted protein degradation. This is highly innovative for use against viral targets, and the present work provides proof of concept for the use of this strategy for the development of novel classes of antivirals.

The authors designed molecules that target the protease/helicase of HCV, using the approved anti-HCV protease drug telaprevir as the basis of the new antivirals. They conjugated telaprevir to ligands that recruit the CRL4CRBN ligase complex. They convincingly showed that their compounds can both inhibit and induce the degradation of the HCV protease. Using elegant controls they demonstrate that protein degradation contributes to its antiviral activity and that the degradation is CRBN-dependent. This work validates a novel strategy for designing antivirals and convincingly serves as the proof of concept for this novel antiviral strategy. While the compounds are slightly less potent than the original antivirals, they remain in the nanomolar range, although they probably depart a bit from the traditional "Lipinsky's rule of five" criteria. Remarkably, evidence is presented that this new class of antiviral agents can be less susceptible to antiviral resistance effects traditional enzymatic inhibitors such as telaprevir.

Overall, strong proof of concept is provided through this innovative work, which is likely to launch new approaches in the design of antiviral strategies.

Few things that need to be addressed:

- As a general comment, while statistics are presented for technical replicates, wherever possible they should also present statistics on the biological replicates (independent experiments).
- The authors should discuss the potential challenge of avoiding the "hook effect" during the course of therapies, where high concentrations of the antivirals are used (many times the EC50 values).
- Provide a reference and a rationale for the statement: "First, targeted protein degraders have been shown to operate at sub-stoichiometric doses and could potentially be less sensitive to point mutations that lead to dramatic losses of affinity and hence potency for traditional occupancy-dependent inhibitors." Is this because the ratio of protein target to "degrader" continuously decreases as the protein is degraded? Please discuss.
- The authors present evidence that the designed degrader analog of telaprevir maintains potency (no significant antiviral resistance) compared to telaprevir. This may be the case for moderate resistance cases such as those shown here (~3-fold and ~10-fold). The authors should discuss what would be expected when significant resistance is observed for the original antiviral. For example, it is not uncommon to have >30-fold or even 100-fold resistance against antivirals through combinations of mutations (often based on steric hindrance or other mechanisms). Comment in discussion whether the resistance profile of the degrader can be significantly improved for such cases where high levels of resistance are observed for the original antiviral component of the degrader.

Response to reviewers:

We thank the reviewers for their careful examination of our manuscript. We believe that we have been able to address the majority of Reviewer #5 concerns and that this has improved the manuscript significantly. Please find below a list of the reviewers' comments, each of which is followed by our response.

REVIEWERS' COMMENTS:**Reviewer #4 (Remarks to the Author):**

The reviewers have satisfied the comments made by Reviewer #2

Reviewer #5 (Remarks to the Author):

This manuscript focuses on an emerging drug approach that relies on targeted protein degradation. This is highly innovative for use against viral targets, and the present work provides proof of concept for the use of this strategy for the development of novel classes of antivirals. The authors designed molecules that target the protease/helicase of HCV, using the approved anti-HCV protease drug telaprevir as the basis of the new antivirals. They conjugated telaprevir to ligands that recruit the CRL4CRBN ligase complex. They convincingly showed that their compounds can both inhibit and induce the degradation of the HCV protease. Using elegant controls they demonstrate that protein degradation contributes to its antiviral activity and that the degradation is CRBN-dependent.

This work validates a novel strategy for designing antivirals and convincingly serves as the proof of concept for this novel antiviral strategy. While the compounds are slightly less potent than the original antivirals, they remain in the nanomolar range, although they probably depart a bit from the traditional "Lipinsky's rule of five" criteria. Remarkably, evidence is presented that this new class of antiviral agents can be less susceptible to antiviral resistance effects traditional enzymatic inhibitors such as telaprevir.

Overall, strong proof of concept is provided through this innovative work, which is likely to launch new approaches in the design of antiviral strategies.

Few things that need to be addressed:

-As a general comment, while statistics are presented for technical replicates, wherever possible they should also present statistics on the biological replicates (independent experiments).

We agree with Reviewer 5 that statistics should be performed on independent experiments, and we believe that our manuscript already complies with this request. As indicated in the figure legends, the data presented in Figure 2d, and in Supplementary Figures 4c and 6c are the average of independent experiments. The p -values for the proteomics data presented in Figure 2b and in Supplementary Figure 3c were calculated using a moderated t-test for three independent biological replicates.

-The authors should discuss the potential challenge of avoiding the “hook effect” during the course of therapies, where high concentrations of the antivirals are used (many times the EC50 values).

Reviewer 5 raises an interesting point. The hook effect is mostly caused by binary interactions that outcompete the formation of the ternary complex that drives target degradation.

With respect to the use of conventional direct-acting antivirals at high concentrations, we note that these agents have occupancy-driven pharmacology and thus require high concentrations to suppress resistance. Our demonstration that telaprevir-resistant HCV mutants are still sensitive to NS3 degraders suggests that the event-driven pharmacology of the degraders may alleviate some of the need for the very high antiviral concentrations *in vivo*. With respect to the telaprevir-based degraders in our study, these compounds illustrate polypharmacology in being able to both inhibit and induce degradation of NS3. Consequently, when concentrations of the degrader exceed those at which ternary complex formation is disfavored, the small molecule can still exert antiviral activity through classical inhibition of protease activity. This is exemplified in our results with compound DGY-08-097. We observed a hook effect when DGY-08-097 was evaluated in the transient NS3 degradation assay at concentrations above 1000 nM (see Figure 2a, and see below); however, DGY-08-097 activity did not plateau when evaluated in the antiviral assays at concentrations above 1000 nM (see Figure 3a and 3c), which is likely explained by an inhibition of HCV protease activity.

We also think that it is worth noting that the hook effect may not be an issue *in vivo*. This is because even if drug exposure is in hook effect range, as the drug gets metabolized it will traverse the concentration range for potent degradation. Consequently, if the time interval for which the drug is in the critical concentration range is sufficient, then target degradation should occur. Of note, there are now two PROTAC degraders from Arvinas in the clinic targeting the estrogen and androgen receptors, and there are multiple examples of small molecule degraders that have a well-documented hook effect in cell culture but still show efficacy in preclinical *in vivo* models of cancer (see for example Zorba et al. Delineating the role of cooperativity in the design

of potent PROTACs for BTK. PNAS 2018). Clearly, more work will be needed to investigate this issue in *in vivo* models of viral infection and to define the therapeutic window for this type of agent.

Last, we note that one way to circumvent the hook effect would be to design bifunctional degraders that have enhanced ternary complex affinity and form specific intermolecular interactions within this complex. Such enhanced cooperativity is expected to counteract the hook effect. Design of such molecules goes well beyond the scope of this study although we are admittedly intrigued by the idea.

We have modified the text in the revised manuscript to discuss this (lines 195-198 in the Results section and lines 270-276 in the Discussion). Line numbers refer to the manuscript with changes tracked with simple markup.

-Provide a reference and a rationale for the statement: “First, targeted protein degraders have been shown to operate at sub-stoichiometric doses and could potentially be less sensitive to point mutations that lead to dramatic losses of affinity and hence potency for traditional occupancy-dependent inhibitors.” Is this because the ratio of protein target to “degrader” continuously decreases as the protein is degraded? Please discuss.

We apologize for not including a reference for this statement. The text was modified to include the following publication:

Bondeson, D. P. *et al.* Catalytic *in vivo* protein knockdown by small-molecule PROTACs. *Nat Chem Biol* 2015, 11(8):611-617, doi:10.1038/nchembio.1858

As stated by the Reviewer, a single degrader molecule can in theory induce ubiquitination of multiple copies of its target; however, the experiments to demonstrate this have generally not been reported. For other degraders developed in our laboratory against kinases and other targets, we observe substoichiometric activity that appears to be due to the event-driven pharmacology, which leads to higher potency than observed for occupancy-driven inhibitors. We modified the text in the revised manuscript to state this (lines 248-249).

-The authors present evidence that the designed degrader analog of telaprevir maintains potency (no significant antiviral resistance) compared to telaprevir. This may be the case for moderate resistance cases such as those shown here (~3-fold and ~10-fold). The authors should discuss what would be expected when significant resistance is observed for the original antiviral. For example, it is not uncommon to have >30-fold or even 100-fold resistance against antivirals through combinations of mutations (often based on steric hindrance or other mechanisms). Comment in discussion whether the resistance profile of the degrader can be significantly improved for such cases where high levels of resistance are observed for the original antiviral component of the degrader.

The Reviewer makes a good point that the mutants evaluated in this study have moderate resistance profiles. As stated above, the remarkable advantage of these small molecules is that their event-driven pharmacology does not require stoichiometric occupancy of the target to have a pharmacological effect. Since they effectively inhibit point mutants with moderate losses of affinity for the target, we believe that treatment with an antiviral degrader will suppress the

evolution of viral variants with the multiple mutations generally observed in viral variants with strong (>30- or 100-fold) resistance to classical inhibitors.

Of note, several studies have demonstrated that small molecule degraders that display lower affinity binding to their target are in fact more potent at promoting target degradation than degraders with higher target affinity. (see for example Tinworth, C. P. et al. PROTAC-Mediated Degradation of Bruton's Tyrosine Kinase Is Inhibited by Covalent Binding. *ACS Chem. Biol.* 2019, 14:342-347). It would therefore be of interest to evaluate whether antiviral degrader activity increases upon the introduction of mutations that reduce affinity and that would confer resistance to conventional inhibitors. We are currently evaluating the resistance profile of the small molecules described in this study and hope to report on our findings in the near future. We modified the text in the revised manuscript to discuss this (lines 265-268).